# Disorder compensation controls doping efficiency in organic semiconductors

Artem Fediai[1], Franz Symalla[2], Pascal Friederich[1,3] & Wolfgang Wenzel [ID] [1]*

Conductivity doping of inorganic and organic semiconductors enables a fantastic variety of highly-efficient electronic devices. While well understood for inorganic materials, the mechanism of doping-induced conductivity and Fermi level shift in organic semiconductors remains elusive. In microscopic simulations with full treatment of many-body Coulomb effects, we reproduce the Fermi level shift in agreement with experimental observations. We find that the additional disorder introduced by doping can actually compensate the intrinsic disorder of the material, such that the total disorder remains constant or is even reduced at doping molar ratios relevant to experiment. In addition to the established dependence of the doping-induced states on the Coulomb interaction in the ionized host-dopant pair, we find that the position of the Fermi level and electrical conductivity is controlled by disorder compensation. By providing a quantitative model for doping in organic semiconductors we enable the predictive design of more efficient redox pairs.

[1] Institute of Nanotechnology, Karlsruhe Institute of Technology, Hermann-von-Helmholtz-Platz 1, 76344 Eggenstein-Leopoldshafen, Germany. [2] Nanomatch GmbH, Hermann-von-Helmholtz-Platz 1, 76344 Eggenstein-Leopoldshafen, Germany. [3] Department of Chemistry, University of Toronto, 80 St George St, Toronto, ON M5S 3H6, Canada. *email: wolfgang.wenzel@kit.edu

Conductivity doping of semiconductors is a ubiquitous technique that is used in most electronic devices[1–3]. The capability of tuning the conductivity type and its magnitude, as well as the Fermi level position by conductivity doping enables all present-day p-n junctions, ohmic contacts to electrodes, field-effect and bipolar transistors. These effects are fully understood for inorganic semiconductors[3]. Molecular doping of organic semiconductors has been pioneered over 20 years ago[4,5], and led to enhanced conductivity and injection properties of organic materials and devices[1,2,6,7]. In contrast to the inorganic universe, our understanding of doping of organic systems lags behind their inorganic counterparts[8–15].

Although the charge carriers generated in organic semiconductors are strongly bound in integer charge transfer complexes[8,13] (ICTC), and their introduction increases the detrimental disorder in the energetic landscape[9], the electrical conductivity upon doping can increase even superlinearly[16] (p-doping), i.e, the average mobility of doping-generated carriers is higher than that in the intrinsic organic semiconductors.

To understand this unusual behavior, we investigated the modification of the density of states, Fermi level shift and the conductivity in a unified theoretical approach, where carriers and their interactions are treated individually with molecular resolution[17]. Our work goes beyond existing approaches and instead of postulating phenomenological modification of the density of states upon doping[9,13,14,18,19]: we extract DOS, Fermi level shift and conductivity from the simulation of the charge carriers dynamics in the actual three-dimensional energy landscape using kinetic Monte-Carlo method[17,20–24] (kMC). Here we analyze the role of energetic disorder in doped organic materials and elucidate the effect of the disorder compensation, which explains the superlinear increase of the conductivity and peculiarities of the Fermi level shift upon doping[16,25–28].

## Results

**Modification of the density of states upon doping.** To compute the relevant elements of the DOS of the doped organic semiconductor, we have performed kinetic Monte-Carlo (kMC) simulations (see Methods and Supplementary Notes 1 and 2, and Supplementary Fig. 1) of a prototypical doped semiconductor ($EA_h = 2.5$ eV, $IP_h = 5.5$ eV, $EA_d = 7$ eV, $IP_d = 9$ eV, where IP and EA denote ionization potential and electron affinity, of the host ($h$) and dopant ($d$), respectively) at small (0.1%), medium (1%), and large (15%) dopant molar ratios (DMRs). The offset energy $E_{off} = EA_d - IP_h = 1.5$ eV assures that the dopant is always ionized (partially ionized dopants will be considered elsewhere, but the effects considered in this work, remain qualitatively the same). Figure 1 shows computed DOS near the Fermi level that features a novel peak comprised by unoccupied states shifted with respect to host HOMO distribution (extraction of the DOS from kMC simulations is described in Supplementary Note 3 and Supplementary Fig. 2). As schematically illustrated in Fig. 2, the novel peak consists mainly of lowest unoccupied molecular orbital (LUMO) states of host cations (denoted by LUMO$^+$) that are part of ICTCs. Note that HOMO and LUMO$^+$ levels are formally introduced as $-IP$ and $-EA^+$, not as the energy levels of a single-particle theory. The two main factors that determine the position of the new peak are the Coulomb interactions in the ICTCs (Fig. 2c) and material disorder (Fig. 2d). Hereafter, we refer to the standard deviation of the HOMO distribution in the intrinsic host material as intrinsic disorder, $\sigma_{int}$. The disorder width in the doped material is called total disorder, $\sigma_{tot}$.

The data in the upper panel of Fig. 1a demonstrate that for small dopant molar ratios, the peak of LUMO$^+$ distribution is separated by the energy of the Coulomb interaction $V_C$ from the original host HOMO level. In addition, we observe the occurrence of the doping-induced disorder: as shown in Fig. 1b, the DOS of the material with zero energetic disorder, where HOMO and LUMO distributions were initially described by $\delta$-functions, broadens into a Gaussian-like distributions. Note that the broadening of the DOS upon doping has been experimentally observed recently[29]. For strong energetic disorder (Fig. 1c), we find that not only the Coulomb interaction, but also the intrinsic disorder changes the peak position of the doping-induced LUMO$^+$ distribution. For $\sigma_{int} = 0.2$ eV (Fig. 1c), the LUMO$^+$ peak is about two times higher with respect to the HOMO as compared to the material with zero intrinsic disorder (Fig. 1b). This can be rationalized as follows: Each dopant in the doped material with intrinsic disorder $\sigma_{int}$ is surrounded by host molecules (six on a cubic lattice). In the absence of energetic disorder, the ionized host-dopant pair (ICTC) will acquire the same stabilizing energy $V_C$ for all six. However, in the case of a disordered material these energies are Gauss distributed with the standard deviation $\sigma_{int}$, which shifts the highest HOMO up by $\approx 2\sigma_{int}$ as compared to the nominal energy level.

**The shift of the Fermi level.** Using the charge neutrality equation, we have determined the position of the Fermi level by simulating a "bulk material" (see Fig. 3a, Supplementary Note 4 and Supplementary Figs. 3 and 4). As illustrated in Fig. 1a, the Fermi level always corresponds to the crossing point between the distributions of the HOMO and the HOMO-derived LUMO$^+$. In Fig. 3b, c we plot the distribution of the DOS, Fermi level $E_F$, peaks of HOMO and LUMO$^+$ and the onset of the HOMO distribution (HOMO$_{onset}$ = $\langle$HOMO$\rangle$ + $2\sigma_{tot}$, where $\langle\ldots\rangle$ denotes the mean) with respect to the vacuum level. The Fermi level lies in the tails of both the electron and hole distributions. This results in a low DOS at a Fermi level and, as a consequence, a relatively low conductivity of the material. Upon doping, the DOS at the Fermi level increases. In case of a material without intrinsic disorder, the DOS at the Fermi energy increases from approximately $5 \times 10^{-5}$ states per eV to $\sim 5 \times 10^{-3}$ states per eV per molecule as the DMR increases from $10^{-3}$ to 0.15. This is caused by a widening of the HOMO and LUMO$^+$ distributions with increasing DMR, which leads to their stronger overlap at the Fermi energy. Thus, highly doped organic materials resemble semimetals rather than semiconductors.

In materials with low intrinsic disorder, the Fermi level remains nearly constant up to high DMRs (Fig. 3b). The disorder-induced upward shift of the LUMO$^+$ (Fig. 2d) is compensated by the increased intensity of LUMO$^+$ (each dopant adds one state to LUMO$^+$), which tends to shift the crossing point of the LUMO$^+$ and HOMO (i.e., the Fermi level) in the opposite direction, that is downward. As a result, upon doping, the Fermi level does not change significantly with respect to the vacuum level. In contrast, the Fermi level decreases with DMR in materials with high disorder (Fig. 3c). Here, the width of the HOMO distribution changes far less and thus the Fermi level is pushed down to the HOMO due to the increased number of the LUMO$^+$ states. In both cases, the Fermi level approaches the onset of the HOMO, resulting in a lower activation barrier for charges near the HOMO onset. Moreover, at a very high DMR, both HOMO and LUMO$^+$ distributions shift down. This is because generated holes occupy first of all the states in the upper tail of the HOMO: the upper part of the HOMO is therefore disappearing upon doping. As a result, mean and peak values of the remaining HOMO distribution shift downward. The LUMO$^+$ is shifted down because it is mainly formed from the HOMO states shifted up by $V_C$.

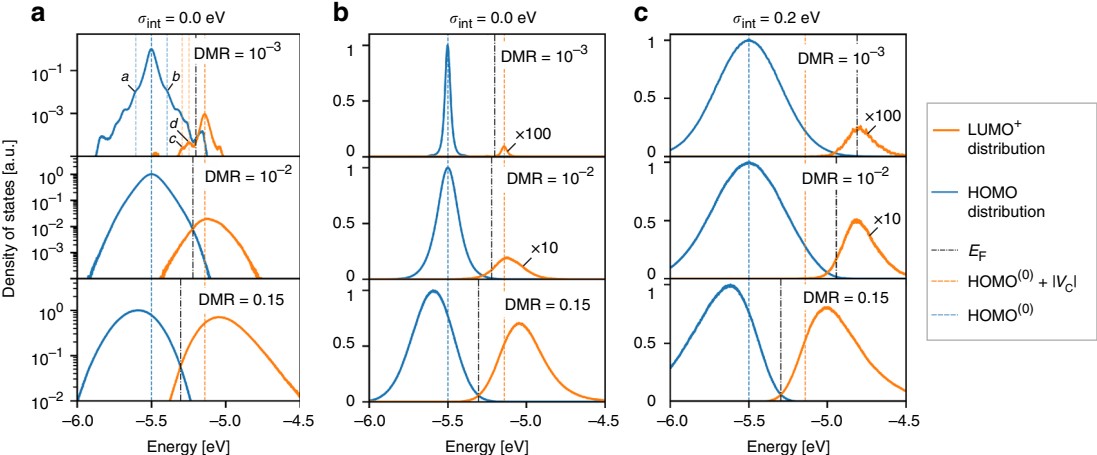

**Fig. 1** Computed density of states (DOS) in doped organic semiconductors. **a** DOS of doped organic semiconductors with zero intrinsic disorder on a logarithmic scale for three dopant molar ratios (DMR) (DMR = {$10^{-3}$, $10^{-2}$, $1.5 \times 10^{-1}$}). Relevant part of the DOS comprises energy distributions of the highest occupied and lowest unoccupied host orbitals, HOMO and LUMO$^+$, respectively. HOMO$^{(0)}$ and $|V_C|$ is the mean HOMO of the corresponding undoped material, and the Coulomb interaction energy between the host cation and dopant anion at a distance $a$ with $a$ being the lattice constant. Feature "$a$" ("$b$") in the top panel corresponds to neutral host molecules at a distance of $a$ to host cations (dopant anions) and $a/\sqrt{2}$ to dopant anions (host cations). Features "$c$" and "$d$" correspond to host cations at distances of $a/\sqrt{3}$ and $a/\sqrt{2}$ to dopant anions, respectively. At low and moderate DMR (top and middle panels), novel LUMO$^+$ distribution appears approximately at HOMO$^{(0)}$ + $|V_C|$. **b** The same as **a** in a linear scale. Note the broadening of HOMO/LUMO$^+$ distributions upon doping. **c** The same as **b** for high ($\sigma_{int} = 0.2$ eV) intrinsic disorder. In this case the mean LUMO$^+$ is separated from HOMO$^{(0)}$ by the energy significantly larger than $|V_C|$ (cf. top panels of **a** and **c**) due to intrinsic disorder. The Fermi level position (denoted by $E_F$) determined from the DOS is always the crossing point of HOMO and LUMO$^+$ distributions so that the density of states at $E_F$ is low. The LUMO$^+$ distribution in panels **b** and **c** has been multiplied by a factor of 100, 10, 1 from top to bottom to enhance visibility at low doping molar rates. The density of states in all panels is normalized so that the maximum DOS value is equal to 1

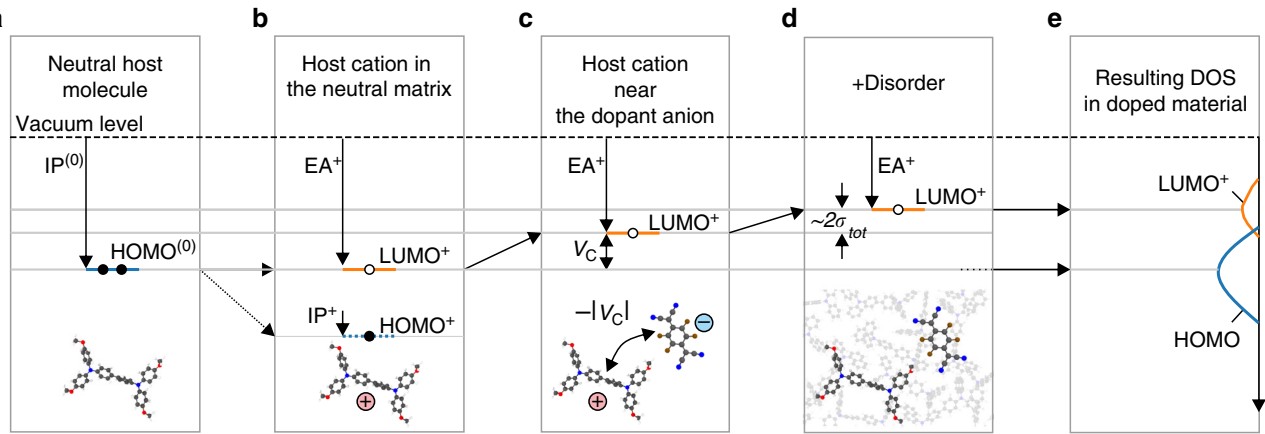

**Fig. 2** Appearance and positioning of the new energy levels upon doping. When a neutral host molecule (**a**) is ionized by donating an electron, its electron affinity, neglecting reorganization effects, will be equal to the ionization potential of the neutral molecule, IP$^{(0)}$ (**b**). In the vicinity of an ionized dopant, the host molecule is part of an integer charge transfer complex (ICTC), which reduces its electron affinity. In case of small total energetic disorder this reduction amounts to the energy of the Coulomb interaction in the ICTC: HOMO − LUMO$^+$ = $V_C = -0.36$ eV (for a distance of 1 nm and $\varepsilon = 4$) (**c**). In presence of disorder, the LUMO$^+$ energy level is shifted up by approximately $2\sigma_{tot}$ (**d**) with $\sigma_{tot}$ being the total energetic disorder, resulting in the distributions in **e**. The second ionization energy of the host, IP$^+$, shown in **b** is assumed to be significantly larger than both IP$^{(0)}$ and the dopant electron affinity, so that it is not relevant for hole transport, and the host molecule cannot be doubly ionized

**Comparison to experiments**. Experimentally the shift of the Fermi level with respect to the onset of the HOMO, $\Delta E_F^{onset}$, is determined from ultraviolet spectroscopy (UPS) measurements of thin (5–20 nm) organic films[25–27]. In order to account for the influence of finite sample thickness and metallic electrodes that determine the global Fermi level, we simulated a single-layer device with a geometry similar to those used in experiment (see Methods, Supplementary Notes 2, 4 and 5, and Supplementary Fig. 5). Figure 4a shows $\Delta E_F^{onset}$ determined in the center of a 15-nm thin doped layer sandwiched between two electrodes with a work function of 4.5 eV, corrected by the difference in polarization energy between of the molecules in the surface and in the bulk[14]. In agreement with experiment[27], we observe two regions with significantly different slope $s \equiv d(E_F - \text{HOMO}_{onset})/d\ln(\text{DMR})$: at low DMR, $\Delta E_F^{onset}$ changes with a slope of $s \approx 1$ eV per decade whereas at high DMR, the slope is only $\approx 0.025$ eV per decade with transition point close to DMR = 0.01 in good agreement with experiments[25–27].

The first (steep) region of the dependence $\Delta E_F^{onset}(\text{DMR})$ disappears in the limit of infinitely thick layer (bulk material, see Supplementary Note 5 and Supplementary Fig. 6). Figure 4b shows the Fermi level shift of simulated bulk systems/thick layers.

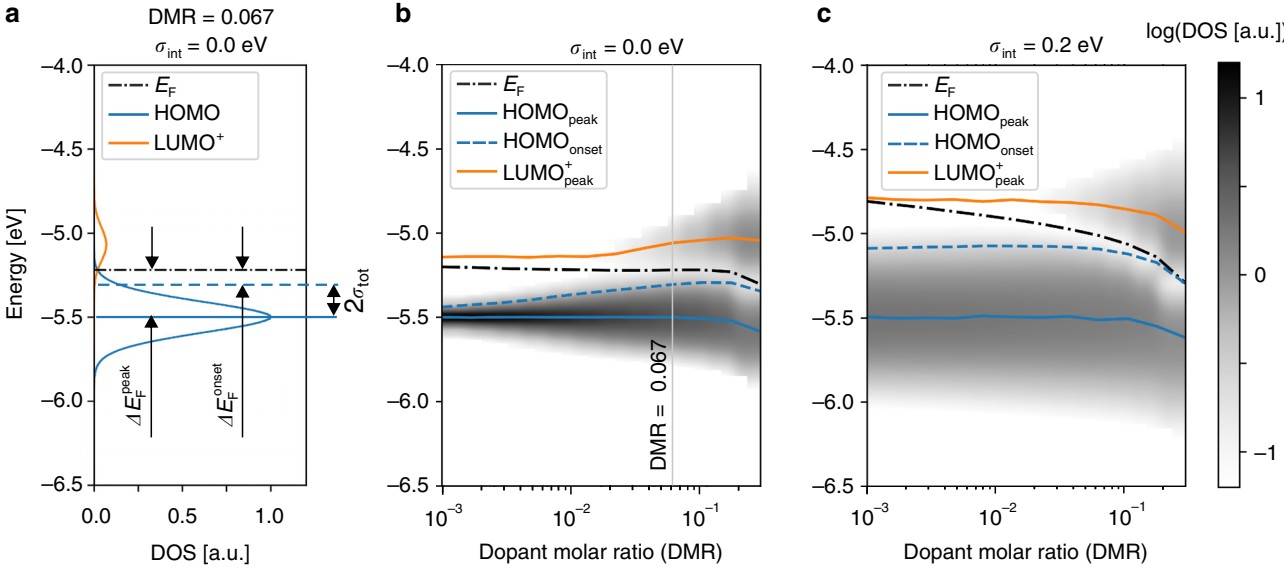

**Fig. 3** Shift of the Fermi level and frontier orbitals energies upon doping. **a** The definition of the Fermi level shift with respect to the peak and the onset of the HOMO, $\Delta E_F^{peak}$ and $\Delta E_F^{onset}$, respectively shown in the material with DMR = 0.067 and zero intrinsic disorder ($\sigma_{int} = 0$). The total material disorder is denoted by $\sigma_{tot}$. **b** and **c**, Evolution of the density of states (DOS) upon doping (the grayscale shows the DOS in logarithmic scale) for organic materials with zero ($\sigma_{int} = 0.0$ eV) and high ($\sigma_{int} = 0.2$ eV) intrinsic disorder, respectively. The peaks of the HOMO and LUMO+ distributions are denoted as HOMO$_{peak}$ and LUMO$_{peak}^+$, respectively. HOMO$_{onset}$ stands for the onset of the HOMO distribution, and $E_F$ is the Fermi level. Panel **a** is a cut of Panel **b** at DMR = 0.067 as shown with a grey line

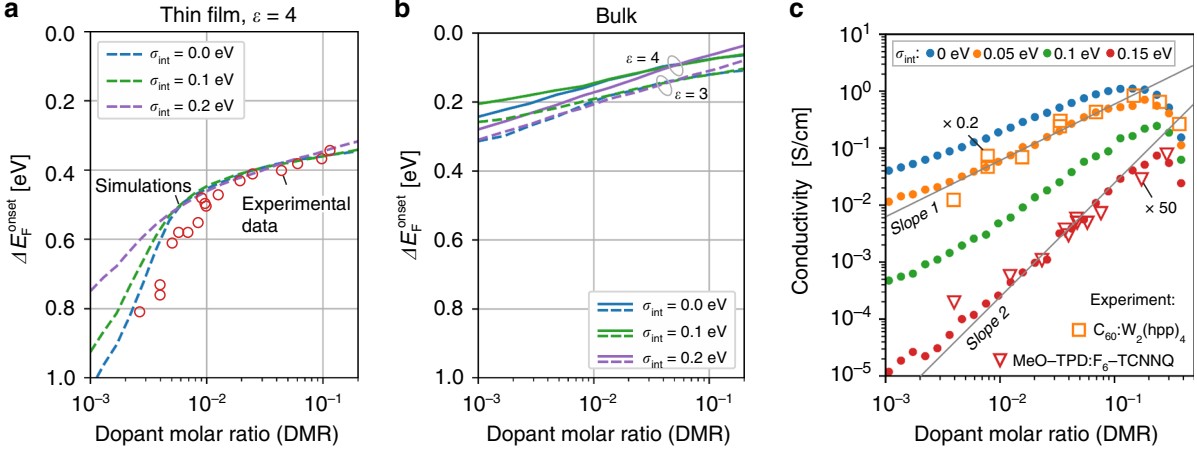

**Fig. 4** Experimental and simulated Fermi level shift and conductivity upon doping. **a** Shift of the Fermi level with respect to the HOMO onset, $\Delta E_F^{onset}$, in the center of a 15 nm doped organic film with $\varepsilon = 4$ and various magnitude of the intrinsic disorder ($\sigma_{int}$) in comparison with UPS measurements[27] of MeO-TPD doped with fluorinated fullerene $C_{60}F_{36}$. Simulated curves are shifted down by 0.3 eV to account for the difference in surface and bulk polarization energies[14]. **b** Shift of the Fermi level with respect to the HOMO onset, $\Delta E_F^{onset}$, computed for a bulk material. **c**, Conductivity as a function of the DMR in doped bulk systems with $\varepsilon = 4$ and various intrinsic disorder strengths in comparison with experimental data[16, 28] for low and highly disordered materials at an electric field strength of 0.04 V nm$^{-1}$. In the moderate doping regime ($10^{-3} <$ DMR $< 10^{-1}$), the conductivity increases either sublinearly (low disorder) or superlinearly (high disorder) upon doping

The Fermi level shift shown in this figure is characteristic for doped bulk materials in contrast to the Fermi level shift calculated in single-layer device simulations or measured in (surface sensitive) UPS experiments. Thus, the following discussion focuses on simulations of bulk materials rather than thin films and single-layer devices.

Using the same kMC protocol, we have computed the doping dependence of the conductivity for materials with varying intrinsic disorder strength (Fig. 4c), which has been computed for a bulk material at an electric field strength of 0.04 V nm$^{-1}$. We found that the conductivity increases either sublinearly (low disorder) or superlinearly (high disorder) upon doping. Here, the

super- and sublinear dependencies refer to the average slope of conductivity as a function of doping in the double logarithmic plot within DMR interval from $10^{-3}$ to $10^{-1}$. For zero disorder, this slope is slightly lower than one (sublinear dependence), whereas for a disorder of 0.15 eV it almost reaches a quadratic dependence. It is obvious, however, that the conductivity in all cases is not exactly a straight line in the log–log scale, that is, it cannot be described by a monomial function. This agrees with the experimental data for $C_{60}$ doped with Tetrakis(hexahydropyrimidinopyrimidine)ditungsten(II) (W$_2$(hpp)$_4$)[16], and N,N,N,N-tetrakis(4-methoxyphenyl)-benzidine (MeO-TPD) doped with 1,3,4,5,7,8-hexafluoro-tetracyanonaphthoquinodimethane

($F_6$TCNNQ)[28], which are representatives of materials with low and high intrinsic disorder, respectively. To explain this behavior we first need to consider the effect of doping-induced disorder depending on the intrinsic disorder of the host material.

**The effect of disorder compensation.** Energetic disorder is the most important material property of amorphous organic semiconductors that determines charge transport[9,21,30]. In undoped samples, requirements on material purity are high, because any kind of impurity typically broadens the density of states and generates trap states for charges. Figure 3b and c reveals significant differences in the broadening of the DOS for semiconductors with low and high intrinsic disorder upon doping. Comparing the width of the HOMO distributions, we note that for low-intrinsic disorder (Fig. 3b), it broadens significantly upon doping. In contrast, for materials with high intrinsic disorder, HOMO distribution remains almost constant. This effect cannot be explained by any model that assumes doping-induced and intrinsic material disorder are uncorrelated. It is the key to understanding, why $p$-doped real-world materials may show the dramatically improved mobility observed in experiment[15].

Figure 5a shows the total disorder $\sigma_{tot}$ (standard deviation of the HOMO distribution) of host molecules in doped materials with various intrinsic disorder strengths $\sigma_{int}$. As expected, $\sigma_{tot}$ is higher in materials with higher $\sigma_{int}$ but the additional energetic disorder defined as $\sigma_{tot} - \sigma_{int}$ shown in Fig. 5b is much smaller in materials with a high intrinsic disorder. This trend is also preserved if we determine the disorder from both HOMO and LUMO$^+$ levels (see Supplementary Note 6 and Supplementary Fig. 7). For a DMR of 10%, the additional energy disorder in a low-intrinsic-disorder material ($\sigma_{int} = 0$) is as high as 110 meV, whereas it amounts to only 12 meV in a material with high intrinsic disorder (0.2 eV). In the limit of high intrinsic disorder and high doping concentrations, we find a constant or even decreasing total disorder, indicating the existence of a correlation between the intrinsic and doping-induced disorder. In other words, the introduction of dopants overcompensates the intrinsic disorder in the material.

For uncorrelated disorder contributions, the doping-induced disorder has to be equal to $(\sigma_{tot}^2 - \sigma_{int}^2)^{1/2}$ for all values of the intrinsic disorder. Figure 5c demonstrates that the doping-induced disorder behaves drastically different: At high doping concentrations, we find that the total disorder is much lower than would be expected on the basis of the results for low-intrinsic disorder. This key result of our investigation can be explained as follows: At high dopant concentrations, holes start filling the upper tail of the host DOS, which reduces the disorder. More precisely, if a dopant molecule in a highly disordered material will produce a hole, the hole will spend most of the time trapped on one of the nearest molecules with the lowest IP. Upon inserting more and more dopants, holes will fill the deepest traps and the material effectively will become less disordered. We call this effect the disorder compensation. It is a reason why, contrary to our intuition, conductivity of highly disordered organic semiconductors increases superlinearly upon doping (Fig. 4c).

We would like to emphasize that computing the energy disorder using only the distribution of free states for holes (HOMO distribution) and not both HOMO and LUMO$^+$ are based on the following considerations. Hole transport is defined mainly by the distribution of the tail states of the IP distribution and the energy of holes that are able to hop, and this latter energy is around the position of the Fermi level. The energy states available for the hop of a hole are only IP states, not EA$^+$ states. As far as the dependence of the Fermi level on the doping is very similar for all intrinsic disorder values (Fig. 4b), disorder of the IP level is what differentiates the relevant energy landscape, in which holes are hopping depending on the intrinsic material disorder.

Above we have considered pure doped organic semiconductors. However, in real organic semiconductors[18,31,32] traps states with concentrations $10^{16}$–$10^{19}$ cm$^{-3}$ are often prevalent. To check how these trap states influence the disorder-compensation effect, we repeated the same simulations for systems that contain

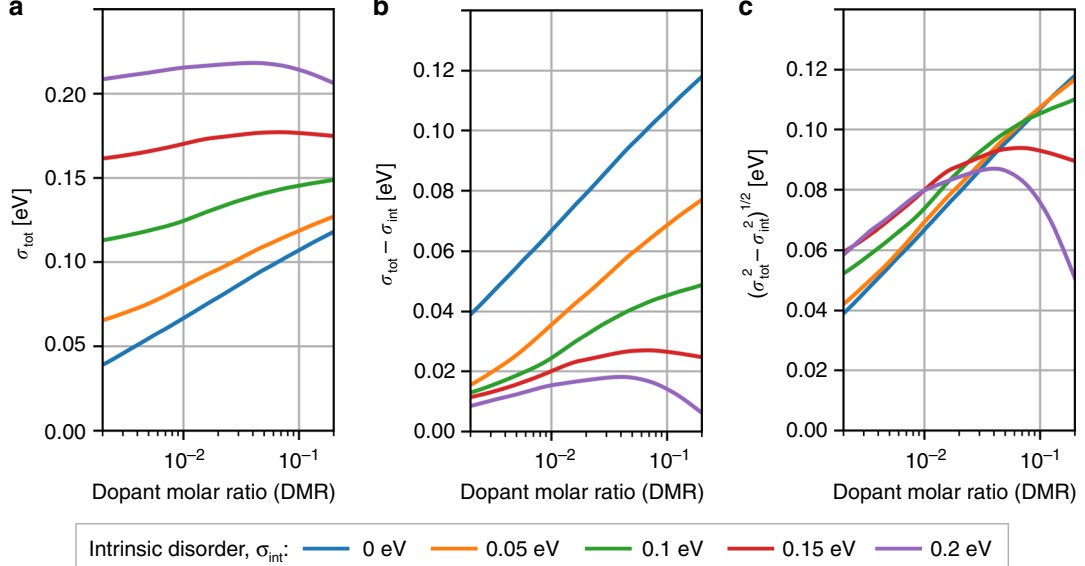

**Fig. 5** Correlation of the intrinsic and doping-induced disorder. **a** Standard deviation of the HOMO distribution (total disorder) $\sigma_{tot}$ of the of host molecules in the doped material as a function of the dopant molar ratio (DMR) for materials with a various intrinsic disorder, $\sigma_{int}$. **b** Excess disorder due to doping, $\sigma_{tot} - \sigma_{int}$. **c** Extrinsic disorder if intrinsic and extrinsic disorder would be not correlated, $(\sigma_{tot}^2 - \sigma_{int}^2)^{1/2}$. Although the total disorder is always larger in materials with stronger intrinsic disorder (**a**), excess disorder due to doping defined as $\sigma_{tot} - \sigma_{int}$ shows an opposite trend: the larger the intrinsic disorder is, the smaller is the excess disorder (**b**). Moreover, the excess disorder may even decrease upon doping of materials with large $\sigma_{int}$. We refer to this effect as the disorder compensation. Panel **c** shows that the doping-induced and intrinsic disorder sources are not fully uncorrelated, otherwise all lines would coincide

trap states (see Supplementary Note 7 and Supplementary Fig. 8). Our simulations revealed that trap filling is the dominant effect at ultralow dopant molar ratios ($DMR < 10^{-3}$), whereas the disorder-compensation effect determines the slope of the conductivity increase in a moderate doping regime ($10^{-3} < DMR < 10^{-1}$).

Extrinsic trap filling reported in the literature and the effect reported here are different not only in terms of doping regimes where they are observed. The disorder-compensation effect means that the increase of the electrostatic disorder due to doping is compensated by the filling of the deepest Coulomb traps, which tends to decrease the total material disorder. It occurs when Coulomb traps have different depth, which is only possible in intrinsically disordered materials. Then a hole fills the deep state of the intrinsic material and the Coulomb trap due to dopant anion. As a result, the total energetic disorder in materials with already high intrinsic disorder may stay the same or even decrease upon doping. In its turn, the effect of the extrinsic trap filling is the filling of deep or shallow extrinsic states due to impurities, rather than (ionized) dopant molecules. The traps filled in this latter case are not of Coulombic nature.

## Discussion

To summarize, the computation of the DOS of doped organic semiconductors by explicit simulation of carriers and their interactions at a microscopic and fully correlated level demonstrates that the observed modification of the DOS does not need the postulation of impurity-induced trap states as argued in earlier models[18] to explain the experimentally observed Fermi level shifts upon doping[25–27] (Fig. 4b). More recent models assumed that doping-induced intergap states are caused by ICTCs, which are separated from the reference energy level (HOMO) by the average Coulomb interaction of the CT state[13,14]. Here we show that in addition to the Coulomb interaction in ICTCs, additional shifts of the same order of magnitude arise due to the entropy and energetic disorder (Fig. 1c), which influences the position of the doping-induced intergap states. Finally, our main finding changes the general view on the interplay of intrinsic and doping-induced disorder that were previously assumed to be uncorrelated[14]. Our simulations demonstrate that intrinsic and doping-induced disorder cannot be considered as separate uncorrelated effects in organic semiconductors, but modify the DOS as the result of a complex interplay of carriers in their energetic landscape (Fig. 5a–c). This disorder-compensation effect explains the observed increase of the average carrier mobility in organic semiconductors[16], despite the introduction of doping as another source of energetic disorder. Note that this superlinear conductivity trend may be explained by two scenarios: either all carriers become more mobile on average, or the fraction of the mobile carriers increases.

Due to disorder compensation, the conductivity of doped organic materials is much less sensitive to the degree of intrinsic disorder of the host material (see Fig. 4c). This significantly broadens the materials space of appropriate host molecules and thus enables the design of novel doped hole/electron transport layers, which are ubiquitously used in most organic electronics devices, ranging from highly-efficient organic light-emitting diodes to organic transistors.

## Methods

**Types of simulated systems.** Two types of the systems have been simulated using kinetic Monte-Carlo method: "bulk semiconductor" and "single-layer device". "Bulk semiconductor" is a simple cubic lattice with $N_x \times N_y \times N_z$ sites and a lattice constant of $a = 1$ nm. Each site represents the center of a mass of an organic molecule (either host or dopant). Periodic boundary conditions (PBCs) are applied in all spatial directions, $x$, $y$, and $z$. "Single-layer device" is a bulk system described above sandwiched between two electrodes. It differs from the first system in several

regards: first, PBCs are applied only in $y$ and $z$ directions; second, each electrode is modelled by $N_y \times N_z$ square lattice sites attached to the bulk semiconductor system from the left and right at a distance of 1 nm. Electrostatic interactions are taken into account using the Ewald summation method[33] as described in Casalegno et al.[34]. Electrodes are assumed to be a perfect metal, therefore, mirror charges are created by all particles in the system.

**Details of kinetic Monte-Carlo simulations.** We consider a model of $p$-doped organic semiconductor at 300 K with relative permittivity $\varepsilon = 4$ where molecular sites are mapped to a cubic lattice with a lattice spacing and electron/hole localization radius 1 nm. Dopant molecules randomly substitute host molecules. Our model includes the following microscopic processes: hopping of charges between two adjacent molecules of one kind, dopant ionization and reciprocal process of dopant neutralization; injection of polarons from electrodes and a reciprocal process of the polaron ejection, all of which are described by Miller-Abrahams rates[35] with an attempt frequency $2 \times 10^9$ sec$^{-1}$ (see Supplementary Note 1 and 2). In simulating electrical conductivity, the field of 0.04 V nm$^{-1}$ is applied.

**Details of numerical implementation.** In all simulations, we have used 50 replicas for each set of material/simulation parameters, and results that we present are averages over these replicas. "Critical" sizes of a replica $N_x \times N_y \times N_z$ that is the minimal sizes that yield the same results as systems with larger sizes have been found for three dopant molar ratios ($10^{-3}$, $10^{-2}$, $10^{-1}$) and each disorder magnitude and then extrapolated for the rest of simulations. To control the convergence upon increasing the system size, we used mean and standard deviation of the HOMO and LUMO$^+$ distributions. As a result we have found that systems with sizes up to $150 \times 150 \times 150$ sites has to be used for low dopant concentration ($DMR = 10^{-3}$) and only down to $20 \times 20 \times 20$ sites is required for a very high dopant concentrations ($DMR = 0.3$). To calculate equilibrium quantities such as DOS and derived quantities (Fermi level) we have modified kMC method. Namely, in dealing with "bulk semiconductor" systems, we set the inverse localization radius in the Miller-Abrahams equation, $b^{-1}$, to zero and connect each site to its 26 nearest neighbors. This allowed us to achieve equilibrium after $10^5$ kMC steps, as compared to $10^6$–$10^8$ kMC steps if using realistic value of $b$ ($\sim 1$ nm$^{-1}$). Setting rates in this fashion influences the time scale of the kMC simulation, but fulfills the detailed balance principle and thus does not change the equilibrium distribution. This has been explicitly justified.

## Data availability

All the data supporting the findings of this study are available within the article, its Supplementary Information files, or from the corresponding author upon reasonable request.

## Code availability

Upon reasonable request, the authors will provide an academic single-user trial license of LightForge for the purpose to reproduce the results of this paper.

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

## Acknowledgements

A.F. and W.W. received funding from the European Union Horizon 2020 research and innovation programme under grant agreement no. 646176 (EXTMOS). P.F. received funding from the European Union Horizon 2020 research and innovation programme under the Marie Skłodowska-Curie grant agreement No 795206. This work was performed on the supercomputer ForHLR-II funded by the Ministry of Science, Research and the Arts Baden-Württemberg and by the Federal Ministry of Education and Research. We acknowledge support by the KIT-Publication Fund of the Karlsruhe Institute of Technology.

## Author contributions

A.F., P.F., and W.W. conceived and designed the project. F.S., developed and employed the methodology to simulate doped organic materials in LightForge kMC, a multi-purpose kinetic Monte-Carlo package, A.F., did the simulations and extracted the relevant materials properties. All authors contributed to preparation of the manuscript. A.F., P.F., and W.W. wrote the manuscript.

## Competing interests

W.W. holds shares of a KIT spinoff, Nanomatch GmbH, which markets software developed by KIT. The other authors declare no competing interests.
