## [Peer Review File · Nature Communications]

Reviewers' comments:

Reviewer #1 (Remarks to the Author):

The manuscript reports a theoretical investigation of the impact of doping and disorder on the density of states of a molecular semiconductor. The key points are the interplay between the broadening of the DOS with increasing dopant molecular ratio (DMR = 0.1% to 15%) and with intrinsic disorder in the organic semiconductor.

I believe that the manuscript is of general interest, given the current importance of better understanding molecular doping in molecular films. Also, experimental observations of DOS broadening upon p-doping have been made, e.g. Lin et al., Chem. Mat. 28, 2677 (2016), and a formal description of the mechanisms at play would be quite interesting. Yet, the manuscript would not be acceptable until several points have been discussed or clarified.

First, the authors aim at explaining with this new model the shift in Fermi level and rise in conductivity upon doping, based solely on these broadening effects. Relevance to real systems, however, must recognize the fact that deep states, traps (electron or hole traps) exist in these materials and that doping-induced trap filling leading to super-linear increases in conductivity, especially at low DMR, has been solidly documented (see, for example, Zhang et al., PRB, 81, 085201 (2010) or Olthof et al., PRL, 109, 176601 (2012)). The relevance of the current manuscript to the explanation of data obtained on real organic film systems is therefore somewhat limited. The authors are requested to discuss this point.

On pages 4 and 5, the authors describe 2 levels of interest, i.e. the HOMO of the neutral host and the LUMO of the host cation LUMO+. Yet, I suppose that not all host cations are located near a dopant anion, as some of the holes surely escape the Coulomb well (otherwise, conductivity would remain low). Why is the LUMO+ of a host cation far away from the dopant not considered in figures 1 and 2? It should be at the level of the HOMO of the neutral host. By the way, there is a mistake in Figure 2c: it is "dopant anion" rather than "dopant cation".

The argument made on p. 7, lines 2-4, "The disorder-induced upward shift because adding more dopants adds more states to the LUMO+" is really not clear. To the reader, it is not clear whether the extra DOS should push closer to or further from the vacuum level. This needs to be clarified. Similarly, the statement made at the top of p. 8, "Moreover, at a very high DMR, higher ionization probability of host molecules with high HOMO" is really unclear.

There is another error at the top of p. 4. The relevant offset energy should be $E_{Ad} - I_{Ph}$.

In summary, the paper is interesting, although whether it rises to the level of Nat. Comm. is debatable, given that the results are obtained for very ideal systems, do not take into consideration very realistic conditions (i.e. no gap states) and thus do not really explain experimental observations.

Reviewer #2 (Remarks to the Author):

This manuscript succeeds to explain a fingerprint experiment related to doping, whose interpretation remained elusive for organic materials: The origin of shift in the Fermi level upon doping.

According to the authors, the shift the Fermi level upon doping can conclusively be explained by considering the occupation of density of states (DOS) of the doped system, independent whether a thin film or bulk material is concerned.

Specifically, the strict consideration of the Coulomb interaction between all charges, allows the

authors to predict how the DOS broadens upon doping and how this broadened DOS affects the occupation with charge carriers.

The predominant - and unanticipated - finding is that the Coulomb interaction-induced disorder partially compensates the inherent energetic disorder.

A second, inherent and major achievement of the present work is that a comprehensive explanation of the doping behavior can be achieved without considering extrinsic factors (such as unintentionally introduced trap states). This result is a necessary prerequisite to design efficient host:dopant combinations.

The employed simulation and extraction methodology appears to be sound and adequate. Importantly, the authors are able to quantify changes in the state energies due to all Coulomb interactions. Key to achieve this quantification is (i) the full consideration of interactions and (ii) the restriction to the thermal equilibrium situation, i.e., a situation in which the occupation statistics is unambiguously captured with a chemical potential.

Yet, there are a number of minor issues that ought to be addressed:

+ Fig 1. panel a, uppermost graph, labels (c) and (d) of the peaks:

Please define clearly the meaning of "first" and "second" nearest neighbor. Is the first nearest neighbor the nearest neighbor or the next nearest neighbor?

Suppose a "first" nearest neighbor is a nearest neighbor. Then, the highest peak (right of peak labeled (c) rather than peak (c)) ought to emerge from the interaction to the nearest neighbor.

+ p.6: Please clearly define the quantity σ_{tot} and its distinction from σ_{int} at the first encounter.

+ p.8. last sentence:

- This sentence is decisive but difficult to read.

- Upon finishing the paragraph, state that the discussion to follow refers to bulk films.

+ p.9., Figure 4c

- For the conductivity simulations, please specify the electric field used. This is particularly intriguing as the electric field lifts detailed balance. It may not necessarily be justified to scale the spatial coupling via parameter β .

- Please specify whether the calculations refer to thin diodes or bulk films.

+ Regarding the observation that the computed Fermi level is always at the intersection energy of LUMO and HOMO associated DOS. Does the Fermi level strictly hit this intersection point or does the observation hold in good approximation?

The former is not necessarily plausible, because the tail of the LUMO DOS visibly changes in shape.

+ Supp. Material, Section S2:

How is the density of ionized dopants (referred to as DMR), that enters the charge neutrality condition, formally calculated?

As the number of ionized dopants is not necessarily equivalent to the density of dopant sites, the degree of ionization must be formally calculated. For the rest of the manuscript, the quantity DMR refers to density of dopant sites; there DMR does not discriminate between neutral and ionized dopants.

Wording (knot-picking)

+ Note that "superlinear" or "sublinear" increase of conductivity with DMR refers to the increase of $\log(\text{conductivity})$ with $\log(\text{DMR})$, i.e., "sublinear" is clearly not sublinear. For the sake of precision, the affected formulations ought to be refined.

Sincerely,
Karin Zojer

Reviewer #3 (Remarks to the Author):

Fedai et al. reported a theoretical and computational study of doping in organic semiconductors, specifically addressing the effect of dopants concentration on the density of states (DOS), conductivity and Fermi level.

The topic is of broad interest and subject of intense research, mostly motivated by the extensive use of doped materials, e.g. in OLED. Theoretical results are compared to experiments and seem providing a rationale for unexplained phenomena, such as for instance the Fermi level shift upon doping.

This study is based on kinetic Monte Carlo simulations on a cubic lattice with explicit treatment of many-body Coulomb interactions. I shall remark that a similar modelling approach has been applied in recent papers (DOI: 10.1038/s41467-018-03302-z, 10.1103/PhysRevB.93.235203; both cited in the manuscript), which also reported super-linear dependence of conductivity with doping.

In the paper under examination there are, however, substantial claims of novelty, especially regarding the interplay between doping-induced and intrinsic disorder, and the shift of the Fermi level upon doping, which seem to closely match experimental trends.

This manuscript may contain enough novelty and insight to be considered for publication on Nature Comm. I would be available to reconsider a revised version once the authors have addressed the comments below.

1. The authors build most of their conclusions around the concept of DOS, which implicitly assumes a single-particle framework. This is inconsistent with the many-body treatment of the entire system where the energy of a given particle depends on the position of the other $N-1$. I understand that mapping a many-body system to an effective single-particle picture is tremendously convenient, but this is not uniquely defined. The authors should clearly explain how do they define the DOS from many-body simulations. The explanation in the SI is largely unsatisfactory.

My (maybe wrong) understanding is that the authors compute the DOS from KMC trajectory, measuring site energies of neutral host sites as IP, and of the site energies of ionized host sites as the cation EA. Is it just the distribution of these two quantities that makes the DOS discussed throughout the paper?

2. The "HOMO" and "LUMO+" DOSs are both relevant to hole carriers. I wonder what is the rationale for plotting the two DOS separately, I find this inaccurate and misleading. The authors should explain what is the rationale for computing the total energetic disorder from the HOMO DOS and not from the total DOS of hole states. This is a crucial point because the main result of the paper is about the disorder compensation and the energetic disorder of the total DOS of hole states largely increases upon doping both in the presence and in the absence of intrinsic disorder.

3. Very low DOS values are found at the Fermi levels and indeed the authors describe doped

materials as semimetals. This is a strong statement, can the authors comment on the dependence of this result on the approach adopted to compute the DOS (see 1).

4. Please clarify what is the difference between the proposed "disorder compensation" mechanism and trap filling, if any.

5. An energy level diagram with an offset between host IP and dopant EA of -1.5 eV is considered as "prototypical". This value is quite unrealistic considering e.g. recent experimental (DOI: 10.1002/adfm.201703780) and theoretical (DOI: 10.1039/c8mh00921j) estimates for similar materials. With such a large offset the dopants will be always ionized (perhaps even doubly ionized) and dopant sites can be considered as point charges, making the offset an irrelevant parameter. Can the author further comment on this choice and its consequences? Would it be meaningful to present results for a more realistic offset?

6. The superlinear trend of conductivity with doping is described in terms of an increase in the mobility. This conclusion seems to be based on the assumption that the charge carrier density increases linearly with doping. This is a controversial subject since it is currently unclear how the (very small) fraction of mobile doping-induced charges evolves with doping.

Please note that is conceptually different saying that carriers become more mobile and that more carriers become mobile with increasing doping!

The author should either provide evidence for an increase in carriers mobility (something I consider unlikely since doping degrades the connectivity and introduces) or present the two possible scenarios for a superlinear conductivity trend.

Minor comments:

- What the author call HOMO+ (HOMO of the cation) in fig 2b is usually referred to as SOMO, singly occupied MO.

- I recommend to revise the manuscript also from a language style standpoint. The paper structure is clear and logical, yet some concepts are too briefly explained and the presentation could be more suited for a broad audience.

Dear Editor,
Dear Reviewers,

Thank you very much for your criticism and valuable suggestions. We have significantly revised our manuscript to address the issues you have raised during your review. We believe that your suggestions have made our work much better and clearer to the broad audience of Nature Communication. We have addressed your questions one-by-one in the tables below. We also submit a document, where we highlighted all changes in the manuscript text file and Supporting Information.

Best regards,
Wolfgang Wenzel on behalf of authors

Referee #1

No.	Question	Answer
-	The manuscript reports a theoretical investigation of the impact of doping and disorder on the density of states of a molecular semiconductor. The key points are the interplay between the broadening of the DOS with increasing dopant molecular ratio (DMR = 0.1% to 15%) and with intrinsic disorder in the organic semiconductor. I believe that the manuscript is of general interest, given the current importance of better understanding molecular doping in molecular films. Also, experimental observations of DOS broadening upon p-doping have been made, e.g. Lin et al., Chem. Mat. 28, 2677 (2016), and a formal description of the mechanisms at play would be quite interesting. Yet, the manuscript would not be acceptable until several points have been discussed or clarified.	-
1.1.	First, the authors aim at explaining with this new model the shift in Fermi level and rise in conductivity upon doping, based solely on these broadening	We have repeated our simulations for the same systems with trap states included (see the end of the main part and Supplementary Section S6). Our simulations (Figure S8)

	effects. Relevance to real systems, however, must recognize the fact that deep states, traps (electron or hole traps) exist in these materials and that doping-induced trap filling leading to super-linear increases in conductivity, especially at low DMR, has been solidly documented (see, for example, Zhang et al., PRB, 81, 085201 (2010) or Olthof et al., PRL, 109, 176601 (2012)). The relevance of the current manuscript to the explanation of data obtained on real organic film systems is therefore somewhat limited. The authors are requested to discuss this point.	revealed that trap states of the realistic concentration cause the superlinear increase of the conductivity at low dopant molar ratios ($<10^{-3}$), whereas at moderate to high dopant molar ratios ($10^{-3} \dots 10^{-1}$) our disorder compensation effect determines the slope of the conductivity curve. Thus, our conclusions are relevant to real organic systems in the moderate to high doping regime. For lower doping concentrations, the regime of the trap filling is dominant, which we have also considered.
1.2.	On pages 4 and 5, the authors describe 2 levels of interest, i.e. the HOMO of the neutral host and the LUMO of the host cation LUMO+. Yet, I suppose that not all host cations are located near a dopant anion, as some of the holes surely escape the Coulomb well (otherwise, conductivity would remain low). Why is the LUMO+ of a host cation far	The LUMO⁺ as extracted from our kMC simulations and shown in Figure 1 is not a (broadened) energy level of some specific host molecule. This is the time-averaged distribution of LUMO⁺ levels across all host molecules of the doped material (if necessary, please, see the averaging procedure in answer to the question 3.1 and detailed description in new Supporting Section S2). This distribution thus contains all dynamic effects associated with carriers moving between dopants. Figure 2e also shows the schematic distribution of the LUMO⁺ and HOMO level. In contrast, Figures 2a--d show HOMO and LUMO⁺ level of individual host molecules, depending on their charged state and position with respect to the dopant anion (either next to the dopant anion or far away). We understood that the usage of the word “level” (that we used for brevity) were misleading, and we have ensured that every instance of the word “level” is substituted by “levels distribution”, if the distribution is meant. In Figure 1, we have considered all host

	away from the dopant not considered in figures 1 and 2? It should be at the level of the HOMO of the neutral host. By the way, there is a mistake in Figure 2c: it is “dopant anion” rather than “dopant cation”.	cations, both trapped in the Coulomb well of dopant anions and far from it. There are of course some host cations escaped from the Coulomb well, but this number is small, therefore DOS has a peak at the energy higher than HOMO by the energy of CT states (plus disorder contribution) and then it decays sharply. Because we define LUMO+ distribution as the time averaged LUMO of the host cations, and the time host cations live far away from the dopant anions is much smaller than next to the dopant anion, the density of LUMO+ states of at HOMO⁽⁰⁾ energy is very small. But it is non zero and sometimes holes jump there, especially if the voltage is applied. We thank you for pointing out the typo. The typo in Figure 2c has been fixed.
1.3.	The argument made on p. 7, lines 2-4, “The disorder-induced upward shift because adding more dopants adds more states to the LUMO+” is really not clear. To the reader, it is not clear whether the extra DOS should push closer to or further from the vacuum level. This needs to be clarified. Similarly, the statement made at the top of p. 8, “Moreover, at a very high DMR, higher ionization probability of host molecules with high HOMO” is really unclear.	We thank for noticing this really bad formulation. We have reformulated this sentence as follows (page 8): “The disorder-induced upward shift of the LUMO+ (Figure 2d) is compensated by the increased intensity of LUMO+ (each dopant adds one state to LUMO+), which tends to shift the crossing point of the LUMO+ and HOMO (i.e the Fermi level) in the opposite direction, that is downward. As a result, upon doping, the Fermi level does not change significantly with respect to the vacuum level.” We have fixed the second sentence as follows: “Moreover, at a very high DMR, both HOMO and LUMO+ distributions shift down. This is because generated holes occupy first of all the states in the upper tail of the HOMO: the upper part of the HOMO is therefore

		disappearing upon doping. As a result, mean and peak values of the remaining HOMO distribution shift downward. The LUMO ⁺ is shifted down because it is mainly formed from the HOMO states shifted up by V_C .”
1.4.	There is another error at the top of p. 4. The relevant offset energy should be EAd - IPh.	We thank for recognizing this typo. It is now fixed.
-	In summary, the paper is interesting, although whether it rises to the level of Nat. Comm. is debatable, given that the results are obtained for very ideal systems, do not take into consideration very realistic conditions (i.e. no gap states) and thus do not really explain experimental observations.	As mentioned above, we have included trap states into our kMC simulations and quantified their effect not only on the conductivity, but also on the Fermi level shift (Supplementary Section S6), which to our knowledge has never been done before. As expected, trap states are extremely important at ultralow dopant concentrations. On the other hand, the effects addressed in our work are relevant at moderate to high dopant concentrations.

Referee #2

	Questions	Answers
-	This manuscript succeeds to explain a fingerprint experiment related to doping, whose interpretation remained elusive for organic materials: The origin of shift in the Fermi level upon doping. According to the authors, the shift the Fermi level upon doping can conclusively be explained by considering the occupation of density of states (DOS) of the doped system, independent whether a thin film or bulk material is concerned. Specifically, the strict consideration of the Coulomb interaction between all charges, allows the authors to predict how the DOS broadens upon doping and how this broadened DOS affects the occupation with charge carriers.	We thank the referee for her interest and positive evaluation of our work. We will in the following address her comments one by one.

	The predominant - and unanticipated - finding is that the Coulomb interaction-induced disorder partially compensates the inherent energetic disorder. A second, inherent and major achievement of the present work is that a comprehensive explanation of the doping behavior can be achieved without considering extrinsic factors (such as unintentionally introduced trap states). This result is a necessary prerequisite to design efficient host:dopant combinations. The employed simulation and extraction methodology appears to be sound and adequate. Importantly, the authors are able to quantify changes in the state energies due to all Coulomb interactions. Key to achieve this quantification is (i) the full consideration of interactions and (ii) the restriction to the thermal equilibrium situation, i.e., a situation in which the occupation statistics is unambiguously captured with a chemical potential. Yet, there a number of minor issues that ought to be addressed:	
2.1	+ Fig 1. panel a, uppermost graph, labels (c) and (d) of the peaks: Please define clearly the meaning of "first" and "second" nearest neighbor. Is the first nearest neighbor the nearest neighbor or the next nearest neighbor? Suppose a "first" nearest neighbor is a nearest neighbor. Then, the highest peak (right of peak labeled (c) rather than peak (c)) ought to emerge from the interaction to the nearest neighbor.	We thank for noticing this unclear and incomplete explanation. We have reformulated those in terms of the lattice constant a: Feature “a” (“b”) in the top panel corresponds to neutral host molecules at a distance of a to host cations (dopant anions) and $a/\sqrt{2}$ to dopant anions (host cations). Features “c” and “d” correspond to host cations at distances of $a/\sqrt{3}$ and $a/\sqrt{2}$ to dopant anions, respectively with a being the lattice constant.
2.2	+ p.6: Please clearly define the quantity σ_{tot} and its distinction from σ_{int} at the first encounter.	We have added the following paragraph in the beginning of the paper:

		Hereafter, we refer to the standard deviation of the HOMO distribution in the intrinsic host material as intrinsic disorder, σ_{int}. The disorder width in the doped material is called total disorder, σ_{tot}.
2.3	+ p.8. last sentence: - This sentence is decisive but difficult to read. - Upon finishing the paragraph, state that the discussion to follow refers to bulk films.	We thank the referee for indicating the need for clarification. We have reformulated the initial sentence from: The Fermi level shift shown in this figure is characteristic for doped bulk materials rather than the Fermi level shift calculated in single-layer device simulations or measured in (surface sensitive) UPS experiments. to: The Fermi level shift shown in this figure is characteristic for doped bulk materials in contrast to the Fermi level shift calculated in single-layer device simulations or measured in (surface sensitive) UPS experiments. We have added the following sentence: Thus, the following discussion focuses on simulations of bulk materials rather than thin films and single layer devices.
2.4	+ p.9., Figure 4c - For the conductivity simulations, please specify the electric field used. This is particularly intriguing as the electric field lifts detailed balance. It may not necessarily be justified to scale the spatial coupling via parameter beta. - Please specify whether the calculations refer to thin diodes or bulk films.	Thank you for noticing this lack of clarity. - The electric field applied is 0.04 V/nm. - The calculations of conductivity refer to a bulk film. We have provided this information in the manuscript (page 9): “Using the same kMC protocol, we have computed the doping dependence of the conductivity for materials with varying intrinsic disorder strength (Figure 4c), which has been computed for a bulk material at an electric field strength of 0.04 V/nm.”

2.5	+ Regarding the observation that the computed Fermi level is always at the intersection energy of LUMO and HOMO associated DOS. Does the Fermi level strictly hits this intersection point or does the observation hold in good approximation? The former is not necessarily plausible, because the tail of the LUMO DOS visibly changes in shape.	The answer is yes, the Fermi level is always the intersection energy of LUMO and HOMO associated DOS. This is because holes associated with LUMO and electrons associated with HOMO obey Fermi statistics, where Fermi level corresponds to the energy of the state with the 50% occupation probability by either holes and electrons. Therefore, density of LUMO and HOMO states is always the same at this energy. Note that we have computed the position of the LUMO/HOMO distribution with respect to Fermi energy by two independent methods:  1. Charge neutrality equation (Equation S5 of the Supplementary Section S3). 2. Equilibration with the chemical potential of the electrodes. The second method does not require to explicitly postulate that the particles are Fermi distributed, but both provided the same result and put the Fermi level at the HOMO/LUMO intersection, which means that the particles obey Fermi statistics. We have added text and Figure S3 to Supplementary Section S3.1 which shows this for various dopant concentrations.
2.6	+ Supp. Material, Section S2: How is the density of ionized dopants (referred to as DMR), that enters the charge neutrality condition, formally calculated? As the number ionized dopants is not necessarily equivalent to the density of dopant sites, the degree of ionization must be formally calculated. For the rest of the manuscript, the quantity DMR refers to density of dopant sites; there DMR does not discriminate between neutral and ionized dopants.	The DMR is the dopant molar ratio rather than the density of ionized dopants and is defined as: $DMR = N_h/N_d$ where N_h and N_d are number (or number density) of host/dopant molecules in the doped material. In this manuscript we intentionally simulated a very strong dopant with EA higher than the host IP by 1.5 eV, therefore DMR and the molar ratio of ionized dopants are the same. Formally, we have computed time averaged fraction of the ionized dopants during each KMC simulation.

		For all simulations shown here, ionized dopant fraction was 1 within machine accuracy. We have added these explanations into Supplementary Section S1.3, and we have also modified the charge neutrality equation (Supplementary Section S2) so that it can be used for partially ionized dopants. We have considered weak dopants in a separate manuscript in more detail, which is about to be submitted. The disorder compensation effect trivially depends on the ionized dopants fraction.
2.7	Wording (knot-picking) + Note that "superlinear" or "sublinear" increase of conductivity with DMR refers to the increase of log(conductivity) with log(DMR), i.e., "sublinear" is clearly not sublinear. For the sake of precision, the affected formulations ought to be refined. Sincerely, Karin Zojer	We are not sure if we understood the comment correctly, but our logic here is as follows. If we consider the plot of $\log(y)$ vs. $\log(x)$, where y and x are conductivity and DMR in our case, and it looks as a straight line in the logarithmic scale, we can represent this dependence in the form: $\log(y) = a \log(x) + b,$ Where the slope a is the power of the dependence of $y(x)$ in a normal scale: $y = \beta x^a$, where $\beta = e^b$. In our case the apparent dependence of the conductivity vs doping is close to a straight line for all disorders and DMR within $[10^{-3}; 10^{-1}]$ and the average slope of this dependence changes from slightly lower than one to almost two as the disorder increases from 0 to 0.015 eV. Therefore, we use the terms "super-" and "sublinear" to explain the difference in the dependence of the conductivity vs. doping for different disorders. Besides, two grey lines that corresponds to the quadratic and linear dependence are drawn in this figure as a guide for an eye.

		At the same time, we agree that in the log-log scale the dependence is not a perfectly straight line. In any case, we did applied those terms to the dependence of $\log(y)$ vs. $\log(x)$. We assumed that the dependence of $\log(y)$ vs. $\log(x)$ is linear. We agree that the text may still be misleading, and to this end added the clarifying text into the manuscript in the paragraph next to Figure 4.
--	--	---

No.	Question	Answer
	Fediai et al. reported a theoretical and computational study of doping in organic semiconductors, specifically addressing the effect of dopants concentration on the density of states (DOS), conductivity and Fermi level. The topic is of broad interest and subject of intense research, mostly motivated by the extensive use of doped materials, e.g. in OLED. Theoretical results are compared to experiments and seem providing a rationale for unexplained phenomena, such as for instance the Fermi level shift upon doping. This study is based on kinetic Monte Carlo simulations on a cubic lattice with explicit treatment of many-body Coulomb interactions. I shall remark that a similar modelling approach has been applied in recent papers (DOI: 10.1038/s41467-018-03302-z, 10.1103/PhysRevB.93.235203; both cited in the manuscript), which also reported super-linear dependence of conductivity with doping. In the paper under examination there are, however, substantial claims of novelty, especially regarding the interplay between doping-induced and intrinsic disorder, and the shift of the Fermi level upon doping, which seem to closely match experimental trends. This manuscript may contain enough novelty and insight to be considered for publication on Nature Comm. I would be available to reconsider a revised version once the authors have addressed the comments below.	-
3.1	1. The authors build most of their conclusions around the concept of DOS, which implicitly assumes a	We thank for noticing that we have not included in clear and detailed description of the density of states into manuscript.

single-particle framework. This is inconsistent with the many-body treatment of the entire system where the energy of a given particle depends on the position of the other $N-1$. I understand that mapping a many-body system to an effective single-particle picture is tremendously convenient, but this is not uniquely defined. The authors should clearly explain how do they define the DOS from many-body simulations. The explanation in the SI is largely unsatisfactory. -- My (maybe wrong) understanding is that the authors compute the DOS from KMC trajectory, measuring site energies of neutral host sites as IP, and of the site energies of ionized host sites as the cation EA. Is it just the distribution of these two quantities that makes the DOS discussed throughout the paper?	In short, the guess of the referee is correct: we computed the DOS from kMC trajectory measuring site energies of neutral hosts as IP and the site energies of ionized hosts as EA+. The DOS of the occupied and free states are the sum of those two quantities. Specifically, at each i-th frame of the kMC trajectory we compute a binned distribution of the energy levels of the neutral hosts $IP_i(E)$ and the dwelling time τ_i of this i-th system state. Then, we find the time-average of these distributions: $IP(E) = 1/T \sum_i \tau_i IP_i(E),$ where $T = \sum_i \tau_i$. A similar procedure is applied to the EA of the ionized host sites yielding EA+ distribution. We note that such a definition of the DOS takes into account that the energy of a given particle depends on the position of other $N - 1$ particles with N being the total number of particles. To our understanding, this is the maximum information we can retrieve from the kMC trajectory to DOS. We believe that this analysis, yields valuable insights into the nature of doping of organic semiconductors which cannot be obtained by any other means. This pertains especially to reliable information about the position of the doping-induced trap states and the DOS modification, essential for doping efficiency. Because the cost of KMC calculations with explicit consideration of the full Coulomb interactions grows dramatically with the number of charges, such calculations may have not been possible before. We therefore believe this study represents a significant step forward, while we obviously acknowledge that it complements other possible approaches to treat doping either analytically or numerically.
--	---

		There is another possible way to compute DOS from the kMC trajectory. In this approach, one first computes the time-averaged charge density at each site (and thus, time-average external potential). After that, one computes the distribution of the IP of the sites subjected to this average potential, which is simply called density of states. In this case, valuable information about the correlations contained in the kMC trajectory is lost. Such a method is essentially a mean-field method. It is likely that in [10.1103/PhysRevB.93.235203] authors used this method, as they observed no signs of EA⁺ (LUMO⁺). As the response to this question, we have implemented this “mean-field” method into our kMC code as well and compared DOS computed by both methods in Supporting Section S2. Apart of this we describe in detail the procedure of extraction of the HOMO and LUMO⁺ distribution (that together comprises the density of states) in the same supporting section.
3.2.	2. The “HOMO” and “LUMO+” DOSs are both relevant to hole carriers. I wonder what is the rationale for plotting the two DOS separately, I find this inaccurate and misleading. The authors should explain what is the rationale for computing the total energetic disorder from the HOMO DOS and not from the total DOS of hole states. This is a crucial point because the main result of the paper is about the disorder compensation and the energetic disorder of the total DOS of hole states largely increases upon doping both in the presence and in the absence of intrinsic disorder.	We fully agree that both HOMO and LUMO⁺ levels are important to hole carriers. To our opinion, there are good reasons to plot LUMO⁺ and HOMO distribution separately. First, we can visually resolve filled and free sites, the shape and extension of their distribution. We also think that the figures are sufficiently clear for the reader to gain an overall picture. Second, the energy of the intersection of the HOMO and LUMO⁺ distribution is the Fermi level (this issue is addressed in detail in our reply to the second referee, Comment 2.5), which is one of the basic parameters to be controlled by conductivity doping. Hole transport is defined mainly by the distribution of the tail states of the IP distribution and the energy of holes that are able to hop, and this energy is around the position of the Fermi level (the position of the

	Fermi level in the sense takes into account the relevant information about occupied states). The energy states available for the hop of a hole are only IP states, not EA⁺ states. As far as the dependence of the Fermi level on the doping is very similar for materials irrespectively to the intrinsic disorder value (Figure 4b), a disorder of the HOMO distribution is what defines the energy landscape, in which relevant holes are hopping, so that we do not need to consider LUMO⁺ level separately. Another argument is to consider how the energy landscape visible to a hole will be changed as we add more and more dopants. In the case of highly disordered material where the IP distribution is almost the same upon doping, the next hole will live in the same energetic landscape as the previous hole, but at a slightly higher in energy where more states are available. This enhances the mobility (that is the conductivity depends on the doping superlinearly). On the other hand, in the material with low intrinsic disorder, every next hole will live higher in energy, but in a more disordered energy landscape. Higher energetic disorder smears the DOS, so that in spite of the upward shift of the Fermi level, the same number or less states are available for hopping. While these arguments are qualitative, we have also compared the disorder from the total DOS that is the sum of LUMO⁺ and HOMO distributions. We replotted Figure 5a,b for the disorder extracted from the total DOS in the Supporting Section S5 (Figure S7). As for the HOMO disorder, the DOS disorder increases more in materials with lower intrinsic disorder. We can therefore conclude as before that the intrinsic disorder compensates the doping-induced disorder, based on an analysis of the total DOS. The physical explanation of the disorder
--	---

		compensation effect is that in the disordered material a hole generated by a dopant tends to trap into the deepest states that are formed due to the combination doping-induced and intrinsic disorder. As a result, the next hole will be moving in the landscape that has fewer deep states or traps (this effect is similar to the trap passivation). In the material with zero disorder, where only doping-induced traps exist, such trap passivation is not as effective: doping creates more traps than it passivates, and there are fewer deep traps. We have included the above-mentioned arguments into the paper (end of the main part) and, as mentioned to the Supplementary Section S5.
3.3.	3. Very low DOS values are found at the Fermi levels and indeed the authors describe doped materials as semimetals. This is a strong statement, can the authors comment on the dependence of this result on the approach adopted to compute the DOS (see 1).	The doped organic semiconductors (OS) are similar to semimetals in that these have vanishing DOS at the Fermi level. The peculiarity of the doped OS is that the vanishing band-gap is of the Coulombic nature. The approach adopted in this work is described in p. 3.1. If we would use the “mean-field” approach to compute DOS (which we have also implemented as a response to this comment), the “semimetal” character of a doped organic semiconductor would not be visible (see Supplementary section S2).
3.4	4. Please clarify what is the difference between the proposed “disorder compensation” mechanism and trap filling, if any.	We have explained the difference between those two effects in the manuscript as follows: Extrinsic trap filling reported in the literature and the effect reported here are different not only in terms of doping regimes where they are observed. The disorder compensation effect means that the increase of the electrostatic disorder due to doping is compensated by the filling of the deepest

		Coulomb traps, which tends to decrease the total material disorder. It occurs when Coulomb traps can have different depth, which is only possible in intrinsically disordered materials. Then a hole “in one hop” fills the deep state of the intrinsic material and the Coulomb trap due to dopant anion. As a result, the total energetic disorder in materials with already high intrinsic disorder may stay the same or even decrease upon doping. In contrast , the effect of the extrinsic trap filling is the filling of deep or shallow extrinsic states due to impurities, rather than (ionized) dopant molecules. The traps filled in this latter case are not of Coulombic nature. As mentioned above, these two effects also different in terms of the doping concentration, where each effect plays the role. The range of doping, where disorder-compensation effect occurs corresponds to moderate doping levels (DMR = 10^{-3}...10^{-1}). The trap filling normally occurs at the ultralow doping levels (DMR < 10^{-3}). This particular issue has also been addressed in the response to Comment 1.1: we have performed a comparative simulations of doped materials with and without trap states -- see Supplementary Section S6. We may note, that trap levels passivation is also observed in inorganic semiconductors; the effect of the disorder compensation reported here is unique to the organic semiconductors.
3.5	5. An energy level diagram with an offset between host IP and dopant EA of -1.5 eV is considered as “prototypical”. This value is quite unrealistic considering e.g. recent experimental (DOI:	In this work, we have intentionally put the EA 1.5 eV higher than IP just to have 100% ionized dopants and no ionization/de-ionization events. The double ionization of the dopant is not allowed in our

	10.1002/adfm.201703780) and theoretical (DOI: 10.1039/c8mh00921j) estimates for similar materials. With such a large offset the dopants will be always ionized (perhaps even doubly ionized) and dopant sites can be considered as point charges, making the offset an irrelevant parameter. Can the author further comment on this choice and its consequences? Would it be meaningful to present results for a more realistic offset?	code. It is indeed assumed that IP and EA differs not so much. We agree with the referee that weak dopants induce additional effects. Given the length of the manuscript and the number of figures we think it best to address these in a forthcoming publication. The ionized dopant fraction of 100% or close to it (experimentally confirmed) is not untypical for doped organic materials, see for instance Table 1 [doi.org/10.1002/adfm.201703780]. We will thoroughly study the effect of the partial dopant ionization in the separate manuscript, which is about to be submitted, where we found a trivial effect of the partial ionization on the disorder compensation effect. Conclusions we made in a given paper are the same if the dopant is not 100% ionized. Unionized dopants are simply excluded from the carrier transport, and do not contribute to the electrostatic disorder, so that at the low to moderate doping levels the situation will simply look like if there were not 100, but 50 dopants instead (for 50% ionized dopant). We have added a brief comment on this issue in the beginning of the manuscript.
3.6	6. The superlinear trend of conductivity with doping is described in terms of an increase in the mobility. This conclusion seems to be based on the assumption that the charge carrier density increases linearly with doping. This is a controversial subject since it is currently unclear how the (very small) fraction of mobile doping-induced charges evolves with doping. Please note that it is conceptually different saying that carriers become more mobile and that more carriers become mobile with increasing doping! The author should either provide evidence for an increase in carriers	By mobility we understand the quantity formally defined as (it has the sense of the average hole mobility): $\mu = \sigma / (en)$ with e, n, σ being elementary charge, hole density and conductivity. In our case n is the same as the dopant density. This definition of mobility assumes that every hole generated by a dopant is a charge carrier so that the charge carrier density increases linearly with doping. We have explained that the superlinear

	mobility (something I consider unlikely since doping degrades the connectivity and introduces) or present the two possible scenarios for a superlinear conductivity trend.	conductivity trend may be explained by two scenarios: either all carriers become more mobile on average, or the fraction of the mobile carriers increases in the text of the manuscript (Discussion Section)
3.7	Minor comments:  - What the author call HOMO+ (HOMO of the cation) in fig 2b is usually referred to as SOMO, singly occupied MO. - I recommend to revise the manuscript also from a language style standpoint. The paper structure is clear and logical, yet some concepts are too briefly explained and the presentation could be more suited for a broad audience. 	 - LUMO⁺ has been formally defined it as -EA⁺. The energy of the remaining electron is however $-IP^+ < -EA^+$ (Figure 2b) because it is the second ionization energy, the difference is the so-called Hubbard U as has been shown in [DOI: 10.1039/C5MH00023H] experimentally. The concept of the SOMO, which came from the single-particle theories does not look suitable in this case. If we think about -EA⁺ as the single-particle orbital energy, this orbital is unoccupied at all rather than singly-occupied by an electron. We have provided necessary explanations in the beginning of the manuscript.

REVIEWERS' COMMENTS:

Reviewer #1 (Remarks to the Author):

The authors have carefully considered the comments from the referee and made appropriate changes and/or provided valuable explanations.

Reviewer #2 (Remarks to the Author):

In their revision, the authors addressed all of my points adequately. In combination with the response and revisions related to the other reviewers, the current manuscript (i) puts forward a fundamental observation related to molecular doping and (ii) is instructive to read.

Reviewer #3 (Remarks to the Author):

The authors put substantial effort in the revision work. They have satisfactorily addressed my critiques and provided the requested clarifications on their work, especially as for it concerns the calculation of the DOS and its implications on transport in doped semiconductors.

Broad interest and novelty were already acknowledged in my previous report. The paper is exhaustive in the revised form and suitable for publication.

Dear Editor,
Dear Reviewers,

Thank you very much for your criticism and valuable suggestions. We have significantly revised our manuscript to address the issues you have raised during your review. We believe that your suggestions have made our work much better and clearer to the broad audience of Nature Communication. We have addressed your questions one-by-one in the tables below. We also submit a document, where we highlighted all changes in the manuscript text file and Supplementary Information.

Best regards,
Wolfgang Wenzel on behalf of authors

Referee #1

No.	Question	Answer
-	The manuscript reports a theoretical investigation of the impact of doping and disorder on the density of states of a molecular semiconductor. The key points are the interplay between the broadening of the DOS with increasing dopant molecular ratio (DMR = 0.1% to 15%) and with intrinsic disorder in the organic semiconductor. I believe that the manuscript is of general interest, given the current importance of better understanding molecular doping in molecular films. Also, experimental observations of DOS broadening upon p-doping have been made, e.g. Lin et al., Chem. Mat. 28, 2677 (2016), and a formal description of the mechanisms at play would be quite interesting. Yet, the manuscript would not be acceptable until several points have been discussed or clarified.	-
1.1.	First, the authors aim at explaining with this new model the shift in Fermi level and rise in conductivity upon doping, based solely on these broadening	We have repeated our simulations for the same systems with trap states included (see the end of the main part and Supplementary Note 7). Our simulations (Supplementary

	effects. Relevance to real systems, however, must recognize the fact that deep states, traps (electron or hole traps) exist in these materials and that doping-induced trap filling leading to super-linear increases in conductivity, especially at low DMR, has been solidly documented (see, for example, Zhang et al., PRB, 81, 085201 (2010) or Olthof et al., PRL, 109, 176601 (2012)). The relevance of the current manuscript to the explanation of data obtained on real organic film systems is therefore somewhat limited. The authors are requested to discuss this point.	Figure 8) revealed that trap states of the realistic concentration cause the superlinear increase of the conductivity at low dopant molar ratios ($<10^{-3}$), whereas at moderate to high dopant molar ratios ($10^{-3} \dots 10^{-1}$) our disorder compensation effect determines the slope of the conductivity curve. Thus, our conclusions are relevant to real organic systems in the moderate to high doping regime. For lower doping concentrations, the regime of the trap filling is dominant, which we have also considered.
1.2.	On pages 4 and 5, the authors describe 2 levels of interest, i.e. the HOMO of the neutral host and the LUMO of the host cation LUMO+. Yet, I suppose that not all host cations are located near a dopant anion, as some of the holes surely escape the Coulomb well (otherwise, conductivity would remain low). Why is the LUMO+ of a host cation far	The LUMO⁺ as extracted from our kMC simulations and shown in Figure 1 is not a (broadened) energy level of some specific host molecule. This is the time-averaged distribution of LUMO⁺ levels across all host molecules of the doped material (if necessary, please, see the averaging procedure in answer to the question 3.1 and detailed description in new Supplementary Note 3). This distribution thus contains all dynamic effects associated with carriers moving between dopants. Figure 2e also shows the schematic distribution of the LUMO⁺ and HOMO level. In contrast, Figures 2a--d show HOMO and LUMO⁺ level of individual host molecules, depending on their charged state and position with respect to the dopant anion (either next to the dopant anion or far away). We understood that the usage of the word “level” (that we used for brevity) were misleading, and we have ensured that every instance of the word “level” is substituted by “levels distribution”, if the distribution is meant. In Figure 1, we have considered all host

	away from the dopant not considered in figures 1 and 2? It should be at the level of the HOMO of the neutral host. By the way, there is a mistake in Figure 2c: it is “dopant anion” rather than “dopant cation”.	cations, both trapped in the Coulomb well of dopant anions and far from it. There are of course some host cations escaped from the Coulomb well, but this number is small, therefore DOS has a peak at the energy higher than HOMO by the energy of CT states (plus disorder contribution) and then it decays sharply. Because we define LUMO+ distribution as the time averaged LUMO of the host cations, and the time host cations live far away from the dopant anions is much smaller than next to the dopant anion, the density of LUMO+ states of at HOMO⁽⁰⁾ energy is very small. But it is non zero and sometimes holes jump there, especially if the voltage is applied. We thank you for pointing out the typo. The typo in Figure 2c has been fixed.
1.3.	The argument made on p. 7, lines 2-4, “The disorder-induced upward shift because adding more dopants adds more states to the LUMO+” is really not clear. To the reader, it is not clear whether the extra DOS should push closer to or further from the vacuum level. This needs to be clarified. Similarly, the statement made at the top of p. 8, “Moreover, at a very high DMR, higher ionization probability of host molecules with high HOMO” is really unclear.	We thank for noticing this really bad formulation. We have reformulated this sentence as follows (page 8): “The disorder-induced upward shift of the LUMO+ (Figure 2d) is compensated by the increased intensity of LUMO+ (each dopant adds one state to LUMO+), which tends to shift the crossing point of the LUMO+ and HOMO (i.e the Fermi level) in the opposite direction, that is downward. As a result, upon doping, the Fermi level does not change significantly with respect to the vacuum level.” We have fixed the second sentence as follows: “Moreover, at a very high DMR, both HOMO and LUMO+ distributions shift down. This is because generated holes occupy first of all the states in the upper tail of the HOMO: the upper part of the

		HOMO is therefore disappearing upon doping. As a result, mean and peak values of the remaining HOMO distribution shift downward. The LUMO ⁺ is shifted down because it is mainly formed from the HOMO states shifted up by V_C .”
1.4.	There is another error at the top of p. 4. The relevant offset energy should be EAd - IPh.	We thank for recognizing this typo. It is now fixed.
-	In summary, the paper is interesting, although whether it rises to the level of Nat. Comm. is debatable, given that the results are obtained for very ideal systems, do not take into consideration very realistic conditions (i.e. no gap states) and thus do not really explain experimental observations.	As mentioned above, we have included trap states into our kMC simulations and quantified their effect not only on the conductivity, but also on the Fermi level shift (Supplementary Note 7), which to our knowledge has never been done before. As expected, trap states are extremely important at ultralow dopant concentrations. On the other hand, the effects addressed in our work are relevant at moderate to high dopant concentrations.

Referee #2

	Questions	Answers
-	This manuscript succeeds to explain a fingerprint experiment related to doping, whose interpretation remained elusive for organic materials: The origin of shift in the Fermi level upon doping. According to the authors, the shift the Fermi level upon doping can conclusively be explained by considering the occupation of density of states (DOS) of the doped system, independent whether a thin film or bulk material is concerned. Specifically, the strict consideration of the Coulomb interaction between all charges, allows the authors to predict how the DOS broadens upon doping and	We thank the referee for her interest and positive evaluation of our work. We will in the following address her comments one by one.

	how this broadened DOS affects the occupation with charge carriers. The predominant - and unanticipated - finding is that the Coulomb interaction-induced disorder partially compensates the inherent energetic disorder. A second, inherent and major achievement of the present work is that a comprehensive explanation of the doping behavior can be achieved without considering extrinsic factors (such as unintentionally introduced trap states). This result is a necessary prerequisite to design efficient host:dopant combinations. The employed simulation and extraction methodology appears to be sound and adequate. Importantly, the authors are able to quantify changes in the state energies due to all Coulomb interactions. Key to achieve this quantification is (i) the full consideration of interactions and (ii) the restriction to the thermal equilibrium situation, i.e., a situation in which the occupation statistics is unambiguously captured with a chemical potential. Yet, there a number of minor issues that ought to be addressed:	
2.1	+ Fig 1. panel a, uppermost graph, labels (c) and (d) of the peaks: Please define clearly the meaning of "first" and "second" nearest neighbor. Is the first nearest neighbor the nearest neighbor or the next nearest neighbor? Suppose a "first" nearest neighbor is a nearest neighbor. Then, the highest peak (right of peak labeled (c) rather than peak (c)) ought to emerge from the interaction to the nearest neighbor.	We thank for noticing this unclear and incomplete explanation. We have reformulated those in terms of the lattice constant a: Feature "a" ("b") in the top panel corresponds to neutral host molecules at a distance of a to host cations (dopant anions) and $a/\sqrt{2}$ to dopant anions (host cations). Features "c" and "d" correspond to host cations at distances of $a/\sqrt{3}$ and $a/\sqrt{2}$ to dopant anions, respectively with a being the lattice constant.

2.2 .	+ p.6: Please clearly define the quantity σ_{tot} and its distinction from σ_{int} at the first encounter.	We have added the following paragraph in the beginning of the paper: Hereafter, we refer to the standard deviation of the HOMO distribution in the intrinsic host material as intrinsic disorder, σ_{int} . The disorder width in the doped material is called total disorder, σ_{tot} .
2.3 .	+ p.8. last sentence: - This sentence is decisive but difficult to read. - Upon finishing the paragraph, state that the discussion to follow refers to bulk films.	We thank the referee for indicating the need for clarification. We have reformulated the initial sentence from: The Fermi level shift shown in this figure is characteristic for doped bulk materials rather than the Fermi level shift calculated in single-layer device simulations or measured in (surface sensitive) UPS experiments. to: The Fermi level shift shown in this figure is characteristic for doped bulk materials in contrast to the Fermi level shift calculated in single-layer device simulations or measured in (surface sensitive) UPS experiments. We have added the following sentence: Thus, the following discussion focuses on simulations of bulk materials rather than thin films and single layer devices.
2.4 .	+ p.9., Figure 4c - For the conductivity simulations, please specify the electric field used. This is particularly intriguing as the electric field lifts detailed balance. It may not necessarily be justified to scale the spatial coupling via parameter beta. - Please specify whether the calculations refer to thin diodes or bulk films.	Thank you for noticing this lack of clarity. - The electric field applied is 0.04 V/nm. - The calculations of conductivity refer to a bulk film. We have provided this information in the manuscript (page 9): “Using the same kMC protocol, we have computed the doping dependence of the conductivity for materials with varying intrinsic disorder strength (Figure 4c), which

		has been computed for a bulk material at an electric field strength of 0.04 V/nm.”
2.5	+ Regarding the observation that the computed Fermi level is always at the intersection energy of LUMO and HOMO associated DOS. Does the Fermi level strictly hits this intersection point or does the observation hold in good approximation? The former is not necessarily plausible, because the tail of the LUMO DOS visibly changes in shape.	The answer is yes, the Fermi level is always the intersection energy of LUMO and HOMO associated DOS. This is because holes associated with LUMO and electrons associated with HOMO obey Fermi statistics, where Fermi level corresponds to the energy of the state with the 50% occupation probability by either holes and electrons. Therefore, density of LUMO and HOMO states is always the same at this energy. Note that we have computed the position of the LUMO/HOMO distribution with respect to Fermi energy by two independent methods:  1. Charge neutrality equation (Supplementary Equation 5 of the Supplementary Note 4). 2. Equilibration with the chemical potential of the electrodes. The second method does not require to explicitly postulate that the particles are Fermi distributed, but both provided the same result and put the Fermi level at the HOMO/LUMO intersection, which means that the particles obey Fermi statistics. We have added text and Supplementary Figure 3 and Supplementary Note 4 which shows this for various dopant concentrations.
2.6	+ Supp. Material, Section S2: How is the density of ionized dopants (referred to as DMR), that enters the charge neutrality condition, formally calculated? As the number ionized dopants is not necessarily equivalent to the density of dopant sites, the degree of ionization must be formally calculated. For the rest of the manuscript, the quantity DMR refers to density of dopant sites; there DMR does not discriminate between	The DMR is the dopant molar ratio rather than the density of ionized dopants and is defined as: $DMR = N_h/N_d$ where N_h and N_d are number (or number density) of host/dopant molecules in the doped material. In this manuscript we intentionally simulated a very strong dopant with EA higher than the

	neutral and ionized dopants.	host IP by 1.5 eV, therefore DMR and the molar ratio of ionized dopants are the same. Formally, we have computed time averaged fraction of the ionized dopants during each kMC simulation. For all simulations shown here, ionized dopant fraction was 1 within machine accuracy. We have added these explanations into Supplementary Note 2, and we have also modified the charge neutrality equation (Supplementary Note 4) so that it can be used for partially ionized dopants. We have considered weak dopants in a separate manuscript in more detail, which is about to be submitted. The disorder compensation effect trivially depends on the ionized dopants fraction.
2.7	Wording (knit-picking) + Note that "superlinear" or "sublinear" increase of conductivity with DMR refers to the increase of log(conductivity) with log(DMR), i.e., "sublinear" is clearly not sublinear. For the sake of precision, the affected formulations ought to be refined. Sincerely, Karin Zojer	We are not sure if we understood the comment correctly, but our logic here is as follows. If we consider the plot of $\log(y)$ vs. $\log(x)$, where y and x are conductivity and DMR in our case, and it looks as a straight line in the logarithmic scale, we can represent this dependence in the form: $\log(y) = a \log(x) + b,$ Where the slope a is the power of the dependence of $y(x)$ in a normal scale: $y = \beta x^a$, where $\beta = e^b$. In our case the apparent dependence of the conductivity vs doping is close to a straight line for all disorders and DMR within $[10^{-3}; 10^{-1}]$ and the average slope of this dependence changes from slightly lower than one to almost two as the disorder increases from 0 to 0.015 eV. Therefore, we use the terms "super-" and

		“sublinear” to explain the difference in the dependence of the conductivity vs. doping for different disorders. Besides, two grey lines that corresponds to the quadratic and linear dependence are drawn in this figure as a guide for an eye. At the same time, we agree that in the log-log scale the dependence is not a perfectly straight line. In any case, we did applied those terms to the dependence of $\log(y)$ vs. $\log(x)$. We assumed that the dependence of $\log(y)$ vs. $\log(x)$ is linear. We agree that the text may still be misleading, and to this end added the clarifying text into the manuscript in the paragraph next to Figure 4.
--	--	--

No.	Question	Answer
	Fediai et al. reported a theoretical and computational study of doping in organic semiconductors, specifically addressing the effect of dopants concentration on the density of states (DOS), conductivity and Fermi level. The topic is of broad interest and subject of intense research, mostly motivated by the extensive use of doped materials, e.g. in OLED. Theoretical results are compared to experiments and seem providing a rationale for unexplained phenomena, such as for instance the Fermi level shift upon doping. This study is based on kinetic Monte Carlo simulations on a cubic lattice with explicit treatment of many-body Coulomb interactions. I shall remark that a similar modelling approach has been applied in recent papers (DOI: 10.1038/s41467-018-03302-z, 10.1103/PhysRevB.93.235203; both cited in the manuscript), which also reported super-linear dependence of conductivity with doping. In the paper under examination there are, however, substantial claims of novelty, especially regarding the interplay between doping-induced and intrinsic disorder, and the shift of the Fermi level upon doping, which seem to closely match experimental trends. This manuscript may contain enough novelty and insight to be considered for publication on Nature Comm. I would be available to reconsider a revised version once the authors have addressed the comments below.	-
3.1	1. The authors build most of their conclusions around the concept of DOS, which implicitly assumes a	We thank for noticing that we have not included in clear and detailed description of the density of states into manuscript.

single-particle framework. This is inconsistent with the many-body treatment of the entire system where the energy of a given particle depends on the position of the other $N-1$. I understand that mapping a many-body system to an effective single-particle picture is tremendously convenient, but this is not uniquely defined. The authors should clearly explain how do they define the DOS from many-body simulations. The explanation in the SI is largely unsatisfactory. -- My (maybe wrong) understanding is that the authors compute the DOS from KMC trajectory, measuring site energies of neutral host sites as IP, and of the site energies of ionized host sites as the cation EA. Is it just the distribution of these two quantities that makes the DOS discussed throughout the paper?	In short, the guess of the referee is correct: we computed the DOS from kMC trajectory measuring site energies of neutral hosts as IP and the site energies of ionized hosts as EA+. The DOS of the occupied and free states are the sum of those two quantities. Specifically, at each i-th frame of the kMC trajectory we compute a binned distribution of the energy levels of the neutral hosts $IP_i(E)$ and the dwelling time τ_i of this i-th system state. Then, we find the time-average of these distributions: $IP(E) = 1/T \sum_i \tau_i IP_i(E),$ where $T = \sum_i \tau_i$. A similar procedure is applied to the EA of the ionized host sites yielding EA+ distribution. We note that such a definition of the DOS takes into account that the energy of a given particle depends on the position of other $N - 1$ particles with N being the total number of particles. To our understanding, this is the maximum information we can retrieve from the kMC trajectory to DOS. We believe that this analysis, yields valuable insights into the nature of doping of organic semiconductors which cannot be obtained by any other means. This pertains especially to reliable information about the position of the doping-induced trap states and the DOS modification, essential for doping efficiency. Because the cost of KMC calculations with explicit consideration of the full Coulomb interactions grows dramatically with the number of charges, such calculations may have not been possible before. We therefore believe this study represents a significant step forward, while we obviously acknowledge that it complements other possible approaches to treat doping either analytically or numerically.
--	---

		There is another possible way to compute DOS from the kMC trajectory. In this approach, one first computes the time-averaged charge density at each site (and thus, time-average external potential). After that, one computes the distribution of the IP of the sites subjected to this average potential, which is simply called density of states. In this case, valuable information about the correlations contained in the kMC trajectory is lost. Such a method is essentially a mean-field method. It is likely that in [10.1103/PhysRevB.93.235203] authors used this method, as they observed no signs of EA⁺ (LUMO⁺). As the response to this question, we have implemented this “mean-field” method into our kMC code as well and compared DOS computed by both methods in Supplementary Note 3. Apart of this we describe in detail the procedure of extraction of the HOMO and LUMO⁺ distribution (that together comprises the density of states) in the same supporting section.
3.2.	2. The “HOMO” and “LUMO+” DOSs are both relevant to hole carriers. I wonder what is the rationale for plotting the two DOS separately, I find this inaccurate and misleading. The authors should explain what is the rationale for computing the total energetic disorder from the HOMO DOS and not from the total DOS of hole states. This is a crucial point because the main result of the paper is about the disorder compensation and the energetic disorder of the total DOS of hole states largely increases upon doping both in the presence and in the absence of intrinsic disorder.	We fully agree that both HOMO and LUMO⁺ levels are important to hole carriers. To our opinion, there are good reasons to plot LUMO⁺ and HOMO distribution separately. First, we can visually resolve filled and free sites, the shape and extension of their distribution. We also think that the figures are sufficiently clear for the reader to gain an overall picture. Second, the energy of the intersection of the HOMO and LUMO⁺ distribution is the Fermi level (this issue is addressed in detail in our reply to the second referee, Comment 2.5), which is one of the basic parameters to be controlled by conductivity doping. Hole transport is defined mainly by the distribution of the tail states of the IP distribution and the energy of holes that are able to hop, and this energy is around the position of the Fermi level (the position of the

	Fermi level in the sense takes into account the relevant information about occupied states). The energy states available for the hop of a hole are only IP states, not EA⁺ states. As far as the dependence of the Fermi level on the doping is very similar for materials irrespectively to the intrinsic disorder value (Figure 4b), a disorder of the HOMO distribution is what defines the energy landscape, in which relevant holes are hopping, so that we do not need to consider LUMO⁺ level separately. Another argument is to consider how the energy landscape visible to a hole will be changed as we add more and more dopants. In the case of highly disordered material where the IP distribution is almost the same upon doping, the next hole will live in the same energetic landscape as the previous hole, but at a slightly higher in energy where more states are available. This enhances the mobility (that is the conductivity depends on the doping superlinearly). On the other hand, in the material with low intrinsic disorder, every next hole will live higher in energy, but in a more disordered energy landscape. Higher energetic disorder smears the DOS, so that in spite of the upward shift of the Fermi level, the same number or less states are available for hopping. While these arguments are qualitative, we have also compared the disorder from the total DOS that is the sum of LUMO⁺ and HOMO distributions. We replotted Figure 5a,b for the disorder extracted from the total DOS in the Supplementary Note 6 (Supplementary Figure 7). As for the HOMO disorder, the DOS disorder increases more in materials with lower intrinsic disorder. We can therefore conclude as before that the intrinsic disorder compensates the doping-induced disorder, based on an analysis of the total DOS.
--	---

		The physical explanation of the disorder compensation effect is that in the disordered material a hole generated by a dopant tends to trap into the deepest states that are formed due to the combination doping-induced and intrinsic disorder. As a result, the next hole will be moving in the landscape that has fewer deep states or traps (this effect is similar to the trap passivation). In the material with zero disorder, where only doping-induced traps exist, such trap passivation is not as effective: doping creates more traps than it passivates, and there are fewer deep traps. We have included the above-mentioned arguments into the paper (end of the main part) and, as mentioned to the Supplementary Note 6.
3.3.	3. Very low DOS values are found at the Fermi levels and indeed the authors describe doped materials as semimetals. This is a strong statement, can the authors comment on the dependence of this result on the approach adopted to compute the DOS (see 1).	The doped organic semiconductors (OS) are similar to semimetals in that these have vanishing DOS at the Fermi level. The peculiarity of the doped OS is that the vanishing band-gap is of the Coulombic nature. The approach adopted in this work is described in p. 3.1. If we would use the “mean-field” approach to compute DOS (which we have also implemented as a response to this comment), the “semimetal” character of a doped organic semiconductor would not be visible (see Supplementary Note 4).
3.4	4. Please clarify what is the difference between the proposed “disorder compensation” mechanism and trap filling, if any.	We have explained the difference between those two effects in the manuscript as follows: Extrinsic trap filling reported in the literature and the effect reported here are different not only in terms of doping regimes where they are observed. The disorder compensation effect means that the increase of the electrostatic disorder due to doping is compensated by the filling of the deepest

		Coulomb traps, which tends to decrease the total material disorder. It occurs when Coulomb traps can have different depth, which is only possible in intrinsically disordered materials. Then a hole “in one hop” fills the deep state of the intrinsic material and the Coulomb trap due to dopant anion. As a result, the total energetic disorder in materials with already high intrinsic disorder may stay the same or even decrease upon doping. In contrast , the effect of the extrinsic trap filling is the filling of deep or shallow extrinsic states due to impurities, rather than (ionized) dopant molecules. The traps filled in this latter case are not of Coulombic nature. As mentioned above, these two effects also different in terms of the doping concentration, where each effect plays the role. The range of doping, where disorder-compensation effect occurs corresponds to moderate doping levels (DMR = 10^{-3}...10^{-1}). The trap filling normally occurs at the ultralow doping levels (DMR < 10^{-3}). This particular issue has also been addressed in the response to Comment 1.1: we have performed a comparative simulations of doped materials with and without trap states -- see Supplementary Note 7. We may note, that trap levels passivation is also observed in inorganic semiconductors; the effect of the disorder compensation reported here is unique to the organic semiconductors.
3.5	5. An energy level diagram with an offset between host IP and dopant EA of -1.5 eV is considered as “prototypical”. This value is quite unrealistic considering e.g. recent experimental (DOI: 10.1002/adfm.201703780) and	In this work, we have intentionally put the EA 1.5 eV higher than IP just to have 100% ionized dopants and no ionization/de-ionization events. The double ionization of the dopant is not allowed in our code. It is indeed assumes that IP and EA

	theoretical (DOI: 10.1039/c8mh00921j) estimates for similar materials. With such a large offset the dopants will be always ionized (perhaps even doubly ionized) and dopant sites can be considered as point charges, making the offset an irrelevant parameter. Can the author further comment on this choice and its consequences? Would it be meaningful to present results for a more realistic offset?	differs not so much. We agree with the referee that weak dopants induce additional effects. Given the length of the manuscript and the number of figures we think it best to address these in a forthcoming publication. The ionized dopant fraction of 100% or close to it (experimentally confirmed) is not untypical for doped organic materials, see for instance Table 1 [doi.org/10.1002/adfm.201703780]. We will thoroughly studied the effect of the partial dopant ionization if the separate manuscript, which is about to be submitted, where we found trivial effect of the partial ionization on the disorder compensation effect. Conclusion we made in a given paper are the same if the dopant is not 100% ionized. Unionized dopants cites are simply excluded from the carrier transport, and do not contribute to the electrostatic disorder, so that at the low to moderate doping levels the situation will simply look like if there were not 100, but 50 dopants instead (for 50% ionized dopant). We have added a brief comment on this issue in the beginning of the manuscript.
3.6	6. The superlinear trend of conductivity with doping is described in terms of an increase in the mobility. This conclusion seems to be based on the assumption that the charge carrier density increases linearly with doping. This is a controversial subject since it is currently unclear how the (very small) fraction of mobile doping-induced charges evolves with doping. Please note that is conceptually different saying that carriers become more mobile and that more carriers become mobile with increasing doping! The author should either provide evidence for an increase in carriers mobility (something I consider unlikely	By mobility we understand the quantity formally defined as (it has the sense of the average hole mobility): $\mu = \sigma/(en)$ with e, n, σ being elementary charge, hole density and conductivity. In our case n is the same as the dopant density. This definition of mobility assumes that every hole generated by a dopant is a charge carrier so that the charge carrier density increases linearly with doping. We have explained that the superlinear conductivity trend may be explained by two

	since doping degrades the connectivity and introduces) or present the two possible scenarios for a superlinear conductivity trend.	scenarios: either all carriers become more mobile on average, or the fraction of the mobile carriers increases in the text of the manuscript (Discussion Section)
3.7	Minor comments:  - What the author call HOMO+ (HOMO of the cation) in fig 2b is usually referred to as SOMO, singly occupied MO. - I recommend to revise the manuscript also from a language style standpoint. The paper structure is clear and logical, yet some concepts are too briefly explained and the presentation could be more suited for a broad audience. 	- LUMO⁺ has been formally defined it as -EA⁺. The energy of the remaining electron is however -IP⁺<-EA⁺ (Figure 2b) because it is the second ionization energy, the difference is the so-called Hubbard U as has been shown in [DOI: 10.1039/C5MH00023H] experimentally. The concept of the SOMO, which came from the single-particle theories does not look suitable in this case. If we think about -EA⁺ as the single-particle orbital energy, this orbital is unoccupied at all rather than singly-occupied by an electron. We have provided necessary explanations in the beginning of the manuscript.

[final comments of reviewers]

REVIEWERS' COMMENTS:

Reviewer #1 (Remarks to the Author):

The authors have carefully considered the comments from the referee and made appropriate changes and/or provided valuable explanations.

Reviewer #2 (Remarks to the Author):

In their revision, the authors addressed all of my points adequately. In combination with the response and revisions related to the other reviewers, the current manuscript (i) puts forward a fundamental observation related to molecular doping and (ii) is instructive to read.

Reviewer #3 (Remarks to the Author):

The authors put substantial effort in the revision work. They have satisfactorily addressed my critiques and provided the requested clarifications on their work, especially as for it concerns the calculation of the DOS and its implications on transport in doped semiconductors.

Broad interest and novelty were already acknowledged in my previous report. The paper is exhaustive in the revised form and suitable for publication.